# The impact of maternal mood and economic stress during Covid-19 pandemic on infant behaviour: Findings from the cross-sectional UK Covid-19 New Mum Study

**Adriana Vázquez-Vázquez** ● *, **Emeline Rougeaux, Sarah Dib, Mary Fewtrell** ● ‡, **Jonathan C. Wells** ● ‡

Childhood Nutrition Research Centre, Population, Policy & Practice Research and Teaching Programme, UCL Great Ormond Street Institute of Child Health, London, United Kingdom

‡ MF and JCW joint senior authors on this work.
* adriana.vazquez.15@ucl.ac.uk

**Data Availability Statement:** We do not have consent from COVID-19 New Mum Study participants to make the raw data publicly available.

## Abstract

The UK Covid-19 New Mum Study (cross-sectional study) recorded maternal experience during the early stages of the pandemic. Our previous analyses showed that the pandemic and 2020 national lockdown negatively impacted maternal mental health. Here, we describe changes in infant behaviour (crying and fussiness) reported by the mother during the Covid-19 pandemic compared to beforehand, and test whether these changes are associated with maternal variables (mental health, coping, financial insecurity, income and household support). We included only responses of mothers whose infants were born before the pandemic started (n = 2,031). Composite scores for maternal mental health and coping were obtained using principal components analysis. Multivariable logistic regression analysis was used to test whether maternal mood and coping and household financial stresses were associated with changes in infant behaviour considered negative (becoming fussier, crying more). Adjusting for confounders, the odds of the infant being fussier and crying more increased by 52% (OR = 1.52, 95% CI = 1.35;1.72) and 64% (OR = 1.64, 95% CI = 1.38;1.95), respectively, if the mother experienced poorer mental health. If the mother coped better and had more time to focus on her health and interests, the odds of these outcomes decreased by 27% (OR = 0.73, 95% CI = 0.65;0.83) and 23% (OR = 0.77, 95% CI = 0.65;0.91), respectively. Mothers who reported that, during the lockdown, household chores were more equally divided 'to a high extent' had 40% (OR = 0.60, 95% CI = 0.39;0.92) lower odds of reporting that their babies became fussier. Reporting major/moderate impact on food expenses was associated with the infant crying more (OR = 2.52, 95% CI = 1.16;5.50). Our results are consistent with previous studies showing that maternal wellbeing plays a significant role in children's behavioural changes during lockdowns. We need strategies to improve mental health and enable women to develop the skills to maintain resilience and reassure their children in challenging times.

However, justified requests for access to anonymized data can be sent to the UCL Ethics Committee (ethics@ucl.ac.uk), who will review these together with the study Principal Investigator Professor Jonathan Wells.

**Funding:** The authors received no specific funding for this work.

**Competing interests:** The authors have declared that no competing interests exist.

## Introduction

Launched in the United Kingdom on May 27th, 2020, the Covid-19 New Mum Study aimed to capture information on maternal experiences and mood and infant feeding practices and behaviour during the initial and subsequent stages of the Covid-19 pandemic. Our previous analyses have shown that the Covid-19 pandemic and the lockdown impacted maternal experiences, resulting in distress for many women, and the need for better infant feeding support, especially face-to-face support for practical issues [1]. Moreover, we were able to identify that a large proportion of new mothers experienced symptoms of poor mental health (feeling down, lonely, irritable and worried) and that travelling for work, having low income, and reporting moderate/major impact on food, rent and essential expenses predicted poorer mental health [2, 3].

The impacts on maternal mental health may be of particular importance for early child development. Studies have suggested that maternal prenatal and postnatal stress could influence cognitive development and temperament in children [4–8]. Since the Covid-19 outbreak, women have been exposed to a stressful situation of unknown duration that can be influencing their children's behaviours [9–15]. An epidemiological study carried out in China during the Covid-19 pandemic on 12,163 children aged 2–5 years and 17,029 children aged 6–12 years showed that mental illness in mothers was more likely to be associated with greater children's psychosocial problems [11]. Another study carried out in Italy found that higher rates of psychological distress in parents were associated with higher levels of hyperactivity/inattention in their children aged 3–13 years [12]. Also in Italy, a study carried out using a longitudinal design reported that children's emotional and behavioural problems significantly increased during the pandemic compared to beforehand, and that maternal mood (anxiety and depression) moderated this trajectory. Specifically, greater maternal mood symptoms were significantly associated with a greater increase in emotional reactive, anxious-depressed, withdrawn and aggressive symptoms during the lockdown [13]. In Spain, a study that examined the effects of confinement on children (3–12 years) and their families reported that parenting distress in particular triggered child negative outcomes (e.g., conduct problems, hyperactivity) [14].

Most previous research has focused on young children. However, infants merit particular attention as their experiences during this life stage may have effects on the bonding with their mothers, and the quality of this bonding can have long-term implications on their behaviour [16, 17]. For instance, Bianco et al [18] reported that prenatal maternal SARS-CoV-2 infection was not associated with infant temperament, but women who reported greater COVID-related life disruptions rated their infants higher on the negative emotionality IBQ-R subscale at 6 months of age. Moreover, higher maternal ratings of stress, as measured by the PSS at 4 months postpartum, were associated with lower values for infants' positive affectivity/surgency and orienting/regulation capacity. Similarly, Provenzi et al reported that maternal prenatal stress due to the Covid-19 pandemic was associated indirectly with lower positive affectivity [19] and directly with lower regulatory capacity [20] in infants. However, it should be noted that at least one study reported no distinct impact of the pandemic on the typical development of temperament across infancy and early childhood [21].

Following the previous research, it is important to know the parental factors that may exacerbate or attenuate the impact of the Covid-19 pandemic and lockdown measures on infant and child wellbeing. Therefore, using data from the Covid-19 New Mum Study and considering the findings in our previous analysis [1–3], we explored changes reported by the mother in specific behaviours of the infant (crying and fussiness) during the lockdown and/or the pandemic and to identify associations with maternal mental health and financial insecurity (Fig 1). Previous studies have reported that maternal emotional states impact maternal reports/

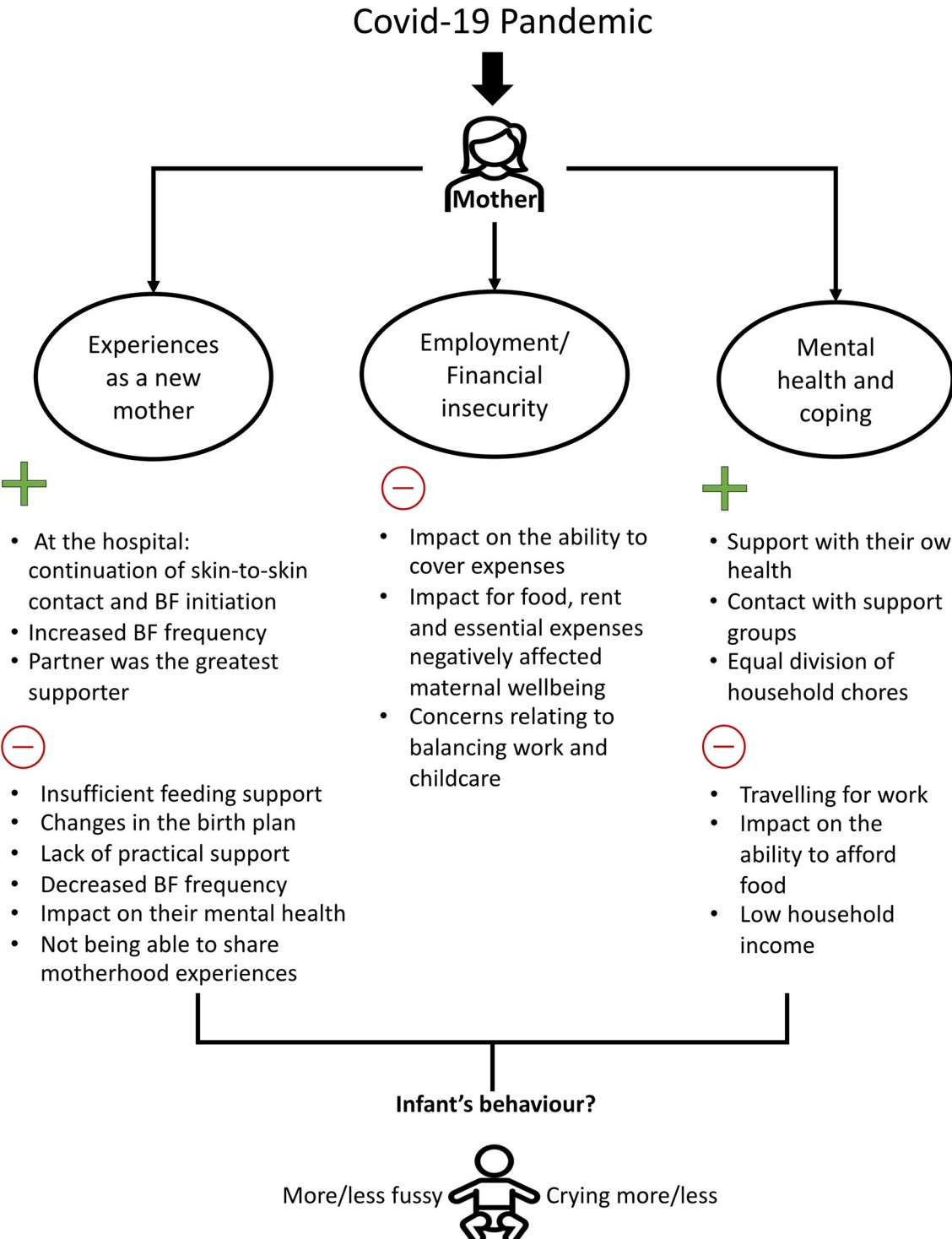

**Fig 1. Positive (+) and negative (-) impacts of the Covid-19 pandemic on new mothers in the Covid-19 New Mum Study.** BF: Breastfeeding. This figure shows the main results of three previous analyses that have been carried out using data from the New Mum Study, carried out in the UK from May 2020 to June 2021. The Covid-19 pandemic and lockdown have affected several aspects of women's lives. As new mothers, they have experienced a lack of support during and after the birth of their babies as well as experienced negative impacts on their mental health [1]. Travelling for work, having a moderate/major impact on the ability to afford food and having a low income predicted poorer mental health [2, 3]. However, on a positive note, greater support with maternal health and with household chores was associated with better mental health and coping [2]. Our current analysis builds on some of these findings to test whether maternal mental health and financial insecurity were associated with changes in infants' crying and fussiness during the Covid-19 pandemic. We hypothesise that greater levels of maternal stressors are associated with a greater likelihood of reporting infants becoming more irritable over time.

perceptions of infant behaviour [22–28], but also these perceptions should be analysed in the context of caregiving. The Covid-19 pandemic and the lockdown caused drastic changes in women's and their infants' lives.

## Methods

Covid-19 was confirmed to be spreading in the UK by the end of January 2020 and this was followed by three national lockdowns. The first was from 23rd of March to 1st of June (when stay-at-home orders were removed) followed by a period of 'relaxed' restrictions until October 2020. The second lockdown came into force on the 5th of November 2020, and was due to end on the 26th of November 2020. Followed by a partial easing of restrictions during December 2020, the UK entered a third lockdown between the 4th of January and 29th of March 2021 (https://www.instituteforgovernment.org.uk/sites/default/files/2022-12/timeline-coronavirus-lockdown-december-2021.pdf). The Covid-19 New Mum Study covered all these lockdown periods and the periods of easing of restrictions until June 2021. However, to analyse changes in infants' behaviour attributable to the pandemic, we included only responses from mothers whose infants were born before the pandemic started. During the study period, a total of 3,430 women resident in the UK responded to the survey, of which 2,031 were from participants whose infants were born before the pandemic. This data was captured between May 2020 and February 2021 (n = 2,031), with the vast majority of the data (98.3%) obtained between the 27th of May and the 28th of August 2020.

Women living in the UK aged ≥18 years who had an infant under 12 months of age were invited to complete the one-time, anonymous online survey generated using RedCap online software. Information and links to the survey were distributed via websites and social media groups used by mothers, such as Facebook infant feeding and mother and baby support groups, Twitter, and Instagram; via existing contacts with relevant professional and support organisations and groups; and via word of mouth.

Ethical approval for the survey was obtained from the UCL Research Ethics Committee (0326/017). The first page of the survey provided information about the study. Having read this and the statements regarding the use of the anonymous data for research purposes, participants were asked to provide consent by ticking a box confirming their willingness to participate and for their data to be analysed by UCL researchers before proceeding with subsequent pages of the survey. They were reminded not to provide any information in the free-text boxes that could allow identification. Despite the anonymity of the data, we also stated that we would use the data collected under the conditions set out by GDPR and UCL University data protection policies. Authors do not have access to information that could identify participants after data collection. Participants could choose not to respond to any questions they felt uncomfortable answering. At the end of the survey, participants were also provided with a list of resources for mental health and infant feeding support, including some tailored to specific groups with protected characteristics.

The full content of the survey, which includes details about the background factors, infant feeding practices, and impacts of Covid-19, has been described elsewhere [1]. Overall, we obtained four types of information: (1) basic social and socio-economic household characteristics; (2) the impact of Covid-19 on household finances and work patterns; (3) infant feeding and behaviour and changes due to Covid-19; and (4) the impact of Covid-19 on the mother's activities, mood and access to support. These data helped identify how Covid-19 was impacting infant feeding practices, which groups were most affected, and which were least able to access relevant support. For the present study, to explore changes in infant behaviour due to

the lockdown and/or the pandemic, we only include the responses provided by mothers whose children were born before 23rd March 2020 (n = 2,031).

## Data

Data captured during the period 27th May 2020 to 6th February 2021 was used for the analysis, and the following variables were included in our analysis:

1. Background characteristics: Social and demographic factors such as maternal age and ethnicity, maternal education and household income, infant's age, sex, gestational age, household status and number of children (< 18 years) living in the household. To avoid bias in terms of background characteristics, a further effort was made to include the experiences of women from ethnically diverse backgrounds and disadvantaged groups to increase the representativeness of our study sample, but little relevant change was attained in our data.

2. Maternal psychosocial wellbeing: markers of Maternal mood (feeling down, feeling lonely, trouble relaxing, becoming easily annoyed or irritable and feeling worried); Coping (having the opportunity to chat with family and friends, enjoying the weather and feeling able to cope with the situation); Behavioural changes (having trouble falling or staying asleep and having a poor appetite); and Time to focus on health and interest (having time to enjoy personal interests, having time to focus on their own health and having time to exercise). These data were further processed as described below to generate composite indices of mental health and coping.

3. Socioeconomic impacts of the pandemic: Measures of household ability to cover expenses (impact on rent/mortgage, food and essential expenses) were included considering that in a recent analysis we found that there is a significantly higher likelihood of having poor psychosocial wellbeing in mothers who reported an impact of the pandemic on their household's ability to cover expenses [3].

4. Household chores. Support with household chores was included, considering that in a previous analysis, we found that more equal division of household chores showed positive associations with maternal mental health and coping [2].

5. Infant behaviour: If the infant was born before the lockdown started, mothers were asked to answer the question, "As a result of staying home or the Covid-19 pandemic, has your baby's behaviour changed?". For this analysis, we focused on changes in irritability, categorised as 'fussy' and 'crying', which the mother could rate as 'more than before', 'the same', or 'less than before'. For data processing and analysis, we combined 'the same' and 'less than before' categories as one group and compared it with the 'more than before' group. All these variables were self-reported by the mother.

## Data processing

Two composite measures of maternal wellbeing were created using Principal Component Analysis (PCA) following the same methodology described in a previous publication [2]. For this analysis, we used 13 variables reflecting maternal psychosocial wellbeing (see previous section). Two components were identified (S1 Table):

1. The first component 'Maternal mental health' incorporated the following traits: feeling down, feeling lonely, trouble relaxing, becoming easily annoyed, feeling worried, having

poor appetite and having trouble falling or staying asleep. For this component, a higher score reflects poorer mental health.

2. The second component 'Coping and focus on health and interests' incorporated the following traits: opportunity to chat with family and friends, enjoying the weather, having time to enjoy personal interests, having time to focus on their own health and having time to exercise. For this component, a higher score reflects better coping and a higher likelihood of focusing on their interests.

The total variation explained by these factors is 49%.

## Statistical analysis

Data from the survey were exported from RedCap and analysed in Stata/SE 15.1. Continuous variables were presented as the mean ± standard deviation (SD) and categorical data by percentages. Logistic regression analysis was used to determine which maternal variables were significantly associated with changes in infants' behaviour (results given as odds ratios [OR] and their 95% CIs and adjusted for confounders [baby´s age, sex and gestational age, ethnicity, maternal education, household status and the total number of children living at the house], using the same group of confounders for all models). We used univariate logistic analysis to test the simple association of each predictor with the outcome. We then conducted multivariate logistic analysis to test the associations when controlling for all other predictors. For this analysis, we focused on negative changes in behaviours (more fussy and more crying than before the pandemic) to identify high-risk groups, i.e., groups that could need more help/support during challenging times such as the pandemic. The models were validated by calculating Hosmer-Lemeshow's goodness-of-fit test. For descriptive purposes, we presented in the tables all the available responses provided per variable, with missing values per variable presented in the footer of the tables. However, for logistic regression, only cases with the completed data required for the analysis were included.

## Results

### Sample characteristics

We identified a total of 2,031 women resident in the UK with infants aged less than 12 months (6.4 ±3.2 months; 49.2% girls) who were born before the lockdown (Table 1). Overall, the mean age of mothers was 32.2 ±4.7 years, and their average total years of education was 16.4 ±3.2 years. Most of these women were of white ethnicity (91.5%), partnered (married, in a civil partnership or cohabitating; 95%) and living in households with a yearly income ≥£45,000 (58.5%).

### Impact of the Covid-19 on infant's behaviour

Of the 2,031 participants, 904 (45.5%) reported changes in their infant's behaviour. Of these 904 mothers, 66% reported that their infant was fussier and 27% that their infant was crying more during the lockdown (Table 2). There were no missing data in this section.

### Predictors of infant's behaviour

Table 2 shows the maternal variables (wellbeing, socio-economic impacts and support with household chores) that are used as predictors of infant behaviour. Univariable logistic analyses of maternal or household economic predictors of infant behaviour are shown in S2 and S3 Tables. Overall, women who experienced negative changes in their mood and difficulties in

**Table 1. Baseline demographic characteristics of mothers and infants (N = 2,031\*).**

| Infant | Mean (± SD) or n (%) |
|---|---|
| Age (months) | 6.4 (2.6) |
| Female sex | 997 (49.2%) |
| Children living at home (Including all ≤12 months) | 1.6 (0.9) |
| **Mother** | |
| Age (years) | 32.2 (4.7) |
| Maternal education (years) | 16.4 (3.2) |
| Household status (%) | |
| Married/civil partnership/cohabitating | 1,904 (95.0%) |
| Single parent, living on own | 68 (3.4%) |
| Single parent, living with family | 33 (1.6%) |
| Household income (yearly, %) | |
| < £20,000 | 139 (6.9%) |
| £20,000–<£30,000 | 201 (10.0%) |
| £30,000–<£45,000 | 377 (18.7%) |
| £45,000–< £75,000 | 630 (31.3%) |
| £75,000–<£100,000 | 289 (14.3%) |
| > £100,000 | 261 (12.9%) |
| Prefer not to say | 113 (5.6%) |
| Ethnicity (%) | |
| White/Caucasian/European | 1842 (91.5%) |
| Asian | 75 (3.6%) |
| Mixed | 59 (2.8%) |
| Black British/African/Caribbean | 23 (1.1%) |
| Other | 12 (1%) |

\*Missing data: Sex = 3; Maternal education = 81; Household status = 26; Household income = 21; Ethnicity = 20.

their ability to cover household expenses had higher odds of reporting difficult infant behaviour during lockdown (S2 Table). Conversely, women who reported being able to cope with the situation and to enjoy personal interests and focus on their own health had lower odds of reporting difficult infant behaviour (S3 Table).

An example of this analysis is shown in Fig 2. Women who reported having trouble relaxing, feeling lonely or experiencing difficulties meeting food expenses had significantly higher odds of reporting that infants were fussier (Fig 2a–2c). In contrast, women who were able to focus on their own health had significantly lower odds of reporting that infants were fussier (Fig 2d).

Multivariable logistic regression analysis revealed, after including all the studied variables (maternal wellbeing PCA components, socioeconomic impacts and support with household chores) and potential confounders (Table 3), that the odds of being fussier and crying more during the pandemic increased by 52% (OR = 1.52, 95% CI = 1.35;1.72, p<0.001) and 64% (OR = 1.64, 95% CI = 1.38;1.95, p<0.001), respectively, if the mother experienced poorer mental health. On the other hand, the odds of being fussier and crying more decreased by 27% (OR = 0.73, 95% CI = 0.65;0.83, p<0.001) and 23% (OR = 0.77, 95% CI = 0.65;0.91, p<0.01), respectively, if the mother coped better and had more time to focus on her health and interests.

**Table 2. The extent to which participant women agreed with the following perceptions.**

| *Infant behaviour* | | | |
|---|---|---|---|
| *Measure*: As a result of staying home or the COVID-19 pandemic, does your baby's behaviour have changed? | | | |
| | More (%) | Less (%) | No change (%) |
| Fussiness | 66.4 | 8.7 | 24.8 |
| Crying | 27.1 | 7.3 | 65.6 |
| *Maternal wellbeing* | | | |
| *Measure*: Since the lockdown, how much do the following statements apply to you? | | | |
| | Not at all (%) | Very little (%) | To some extent (%) | To a high extent (%) |
| Feeling down | 10.4 | 27.6 | 39.0 | 23.0 |
| Lonely | 14.6 | 20.7 | 34.0 | 30.7 |
| Easily annoyed | 10.5 | 23.2 | 37.1 | 29.2 |
| Trouble relaxing | 13.4 | 23.2 | 35.8 | 27.6 |
| Feeling worried | 6.7 | 18.2 | 38.4 | 36.7 |
| Poor appetite | 61.6 | 18.4 | 15.3 | 4.7 |
| Trouble sleeping | 28.8 | 24.9 | 27.0 | 19.3 |
| Opportunity to chat with friends | 1.2 | 10.3 | 43.2 | 45.3 |
| Enjoying the weather | 5.4 | 14.2 | 39.0 | 41.4 |
| Feeling able to cope with the situation | 9.8 | 25.9 | 52.2 | 12.1 |
| Enjoy personal interests | 50.2 | 34.2 | 12.3 | 3.3 |
| Focus on health | 33.8 | 38.9 | 21.1 | 6.1 |
| Time to exercise | 24.5 | 35.1 | 28.7 | 11.7 |
| *Socioeconomic impacts* | | | |
| *Measure*: Which of the following best describes the impact of COVID-19 on your household's ability to pay for rent/food/essential expenses? | | | |
| | No impact (%) | Minor (%) | Moderate/Major (%) |
| Impact on rent | 64.6 | 17.6 | 17.8 |
| Impact on food expenses | 68.1 | 19.0 | 12.9 |
| Impact on essential expenses | 71.2 | 16.7 | 12.1 |
| *Support with household chores* | | | |
| *Measure*: I feel the house chores are more equally divided among household members | | | |
| Not at all | Very little (%) | To some extent (%) | To a high extent (%) |
| 41.7 | 25.1 | 21.8 | 11.4 |

Women who reported that household chores were more equally divided 'to a high extent' had 40% (OR = 0.60, 95% IC = 0.39;0.92, p = 0.02) lower odds of reporting that their babies were fussier. Reporting a minor and moderate/major impact on food expenses was positively associated with the infant crying more (OR = 2.01, 95% CI = 1.15, p = 0.01; 3.52 and OR = 2.52, 95% CI = 1.16;5.50, p = 0.02, respectively). A household income < £30,000 was positively associated with the infant being fussier (OR = 1.99, 95% CI = 1.20;3.34, p<0.01) and women living in households with incomes < £45,000, < £30,000 and < £20,000 had higher odds of reporting that infant was crying more during the lockdown (OR = 1.98, 95% CI = 0.99;3.97, p = 0.05; OR = 2.81, 95% CI = 1.33;5.90, p<0.01; OR = 2.87, 95% CI = 1.23;6.71, p = 0.01, respectively) (Table 3).

## Discussion

The lockdown measures introduced in March 2020 to reduce the spread of Covid-19 abruptly changed the routines and interactions of families in the UK. The present study aimed to explore changes in infant behaviour during the Covid-19 pandemic and analyse associations

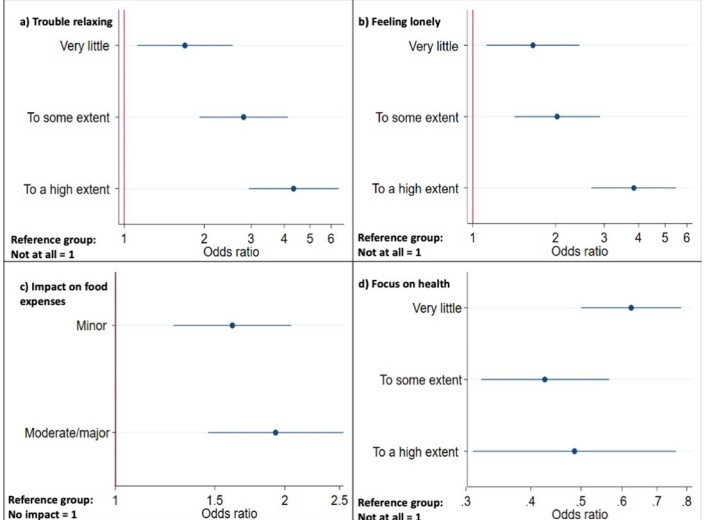

**Fig 2. Univariable logistic regression analysis predicting infant fussiness.** Maternal predictors: a) Trouble relaxing, b) Feeling lonely, c) Perceiving impacts on food expenses and d) Being able to focus on their own health. Women who reported having trouble relaxing, feeling lonely and having an impact on food expenses had significantly higher odds of reporting that infants were fussier (Fig 2a-2c). On the other hand, women who were able to focus on their own health had significantly lower odds of reporting that infants were fussier (Fig 2d).

with maternal-related variables. As stated in the introduction, this analysis builds on previous findings (Fig 1) on maternal mental health using data from the Covid-19 New Mum Study.

It is noteworthy that the results revealed that 54.5% did not report a change in infant behaviour during the lockdown. However, it should also be noted that around 45.5% displayed behavioural changes according to the mothers. Moreover, many of these infants changed in the direction of more difficult behaviour (fussier and crying more), suggesting that the pandemic and confinement could negatively affect their behaviour. These results are in line with some previous studies that show changes in children's behavioural and emotional adjustment during confinement, i.e., higher levels of anxiety and depressive symptoms [29], hyperactive behaviours, showing more restlessness, boredom and difficulties in keeping focused [10], increase of anxious-depressive and aggressive symptoms [30] and lower sleep quality [9].

More interestingly, our analysis agrees with other studies [11–14] that maternal wellbeing plays a significant role in children's behavioural changes during the lockdown. Particularly, we found that poor maternal mental health and having food insecurity and low income increased the odds of reporting more difficult behaviours in their infant. These results extend existing evidence on the role of maternal mental health and children's behavioural adjustment [31] by showing that maternal distress under stressful circumstances, such as the lockdown resulting from the pandemic, could be detrimental to their infant's wellbeing.

In addition, in line with previous studies [13, 14], our results showed the protective role of maternal mental wellbeing during the lockdown on infants behaviour. Being able to cope and to focus on their own health and interests were associated with reduced odds of reporting difficult behaviours in their infants.

Unfortunately, we do not have more detailed information on changes in the behaviour of infants during confinement. Nevertheless, considering our previous findings [1–3], it is possible to hypothesize a potential pathway that may explain this intertwining between maternal mental health and infant behaviour during confinement. In one of our previous reports [3], we

**Table 3. Multivariable logistic regression analysis predicting infant behaviour.**

| Predictors | Model 1 (n = 1,636; p<0.001) Fussier | | | | Model 2 (n = 1,627; p<0.001) Crying more | | | |
|---|---|---|---|---|---|---|---|---|
| | OR | SE | p-value | 95%CI | OR | SE | p-value | 95%CI |
| **Age (months)** | 1.18 | 0.03 | **<0.001** | 1.12;1.23 | 1.06 | 0.03 | **0.037** | 1.00;1.13 |
| **Maternal mental health** | 1.52 | 0.09 | **<0.001** | 1.35;1.72 | 1.64 | 0.14 | **<0.001** | 1.38;1.95 |
| **Coping and focus on health and interests** | 0.73 | 0.05 | **<0.001** | 0.65;0.83 | 0.77 | 0.07 | **<0.01** | 0.65;0.91 |
| **House chores more equally divided** | | | | | | | | |
| Not at all | Reference group | | | | | | | |
| Very little | 0.78 | 0.12 | 0.11 | 0.58;1.05 | 1.04 | 0.21 | 0.81 | 0.70;1.56 |
| To some extent | 0.89 | 0.14 | 0.44 | 0.65;1.20 | 0.84 | 0.19 | 0.43 | 0.54;1.31 |
| To a high extent | 0.60 | 0.13 | **0.02** | 0.39;0.92 | 0.82 | 0.24 | 0.50 | 0.46;1.45 |
| **Impact on expenses** | | | | | | | | |
| *Impact on rent* | | | | | | | | |
| No impact | Reference group | | | | | | | |
| Minor impact | 1.40 | 0.25 | 0.06 | 0.98;1.99 | 0.83 | 0.21 | 0.46 | 0.51;1.36 |
| Moderate/major impact | 1.29 | 0.29 | 0.26 | 0.83;2.02 | 0.70 | 0.21 | 0.25 | 0.39;1.27 |
| *Food expenses* | | | | | | | | |
| No impact | Reference group | | | | | | | |
| Minor impact | 1.23 | 0.27 | 0.33 | 0.80;1.90 | 2.01 | 0.57 | **0.01** | 1.15;3.52 |
| Moderate/major impact | 1.30 | 0.42 | 0.41 | 0.69;2.44 | 2.52 | 1.00 | **0.02** | 1.16;5.50 |
| *Essentials* | | | | | | | | |
| No impact | Reference group | | | | | | | |
| Minor impact | 0.73 | 0.18 | 0.20 | 0.46;1.18 | 0.74 | 0.23 | 0.34 | 0.40;1.37 |
| Moderate/major impact | 0.87 | 0.29 | 0.69 | 0.45;1.69 | 1.05 | 0.43 | 0.91 | 0.47;2.34 |
| **Income** | | | | | | | | |
| >£100,000 | Reference group | | | | | | | |
| £75,000–<£100,000 | 1.14 | 0.27 | 0.59 | 0.71;1.81 | 1.25 | 0.48 | 0.56 | 0.59;2.66 |
| £45,000–<£75,000 | 1.47 | 0.31 | 0.06 | 0.98;2.21 | 1.41 | 0.48 | 0.31 | 0.72;2.76 |
| £30,000–<£45,000 | 1.51 | 0.34 | 0.07 | 0.97;2.35 | 1.98 | 0.70 | **0.05** | 0.99;3.97 |
| £20,000–<£30,000 | 1.99 | 0.52 | **<0.01** | 1.20;3.34 | 2.81 | 1.06 | **<0.01** | 1.33;5.90 |
| < £20,000 | 1.30 | 0.42 | 0.41 | 0.69;2.44 | 2.87 | 1.24 | **0.01** | 1.23;6.71 |

OR: Odds ratio; SE: Standard error; CI: Confidence interval. Models were controlled for the baby's age, sex and gestational age, ethnicity, maternal education, household status and the total number of children living in the house. The models were validated by calculating the Hosmer-Lemeshow's goodness of fit test (Model 1: p = 0.41; Model 2: p = 0.78). Because the tests were not significant, we were satisfied with the fit of our models.

included free text responses from the mothers about the impact of the pandemic on employment. Several mothers reported struggles relating to balancing work and childcare due to school/preschool closures and/or changes in work hours incompatible with childcare. Moreover, they reported experiencing household income reduction and fears of potential redundancy (mostly after the end of maternity leave). Therefore, the negative impacts on mental health, related to a lack of practical support, financial insecurity, and the fear of losing their jobs, could worsen maternal ability and efficacy in dealing with increased caregiving responsibilities related to the stay-at-home request. This could be detrimental to the mother-child relationship and affect the infant's behaviour. A recent study reported that Covid-19 related concerns, particularly grief experiences, pose unique risks to the maternal–infant bonding process [32] and the quality of this bonding can have effects on infant behaviour [16, 17].

Our study has limitations that should be considered, given the complexity of carrying out a study under crises conditions and home confinement. First, the cross-sectional design and that the current population is not representative of all new mothers in the UK. Compared to national data, our participants have higher educational attainment, higher income, are more likely to be married or cohabitating and are more likely to be of white ethnic background. Further effort was made to include the experiences of women from ethnically diverse backgrounds and disadvantaged groups to increase the representativeness of our study sample, but little relevant change was attained in our data. Moreover, we did not specifically survey maternal mental ill-health/depression and temperament in infants. Therefore, we did not formally assess maternal mental health and infant behaviour. Consequently, we were not able to compare the rates of depression or anxiety in new mothers or difficult behaviours in children, such as negative affectivity, during the Covid-19 pandemic with the rates before the pandemic. Through our survey, we wanted to learn the challenges that new mothers experienced as well as how to improve the support available.

In addition, although maternal retrospective recall on infants' behaviour allowed us to explore behavioural changes after exposure to a unique and unpredictable phenomenon, the Covid-19 pandemic, it could also potentially introduce bias in the recollection process. However, most of the data used in our analysis was obtained between the end of May and August 2020 (n = 1,997/2,031), at the beginning of the pandemic. Hence, exposure to the pandemic would have been relatively fresh in the participant's memory. Moreover, how women perceive their mental health could influence their evaluation of their infants' behaviour, e.g., perceiving themselves as having poor mental health could cause mothers to perceive their infants as having negative changes in behaviour, even if this is not the case. However, this could also be seen as a strength of the study. Prior studies have suggested that mood affects women's perceptual accuracy when evaluating their infants' behaviour/mood states. This distorted evaluation may be persistent over time and potentially affect children's cognitive development if the evaluation of the child becomes more negative [22–27]. In fact, our results are in line with a recent study published that women who reported higher perceived stress during the pandemic rated their infants lower in the positive affectivity/surgency and orienting/regulation scales, and those who reported a higher negative impact of the pandemic in their life rated their infants higher in the negative emotionality scale [18]. Finally, this is a cross-sectional survey, and we are unable to infer causality; however, longitudinal research mentioned previously reported that children's emotional and behavioural problems significantly increased from pre- to during the pandemic and that maternal mood (anxiety and depression) moderated this trajectory [13].

Despite these limitations, our study is in line with previous research that suggests that maternal mental wellbeing in times of emergency might detriment or mitigate the adverse impact of the Covid-19 lockdown on children's behavioural adjustment. Our findings highlight the need to establish strategies to improve maternal health and coping behaviours, enabling them to develop the skills to maintain resilience and reassure their children in challenging times.

## Supporting information

**S1 Table. Maternal wellbeing components from the principal component analysis (PCA); N = 2,031.** Kaiser-Meyer-Olkin Measure of Sampling Adequacy = 0.885. Bartlett test of sphericity: p = <0.001. Rotation method: Varimax with Kaiser normalization. Component 1: Higher score reflect poor mental health. Component 2: Higher score reflect better coping. (DOCX)

**S2 Table. Univariable logistic regression analysis predicting infant behaviour.**
(DOCX)

**S3 Table. Univariable logistic regression analysis predicting infant behaviour.**
(DOCX)

## Acknowledgments

All research at Great Ormond Street Hospital NHS Foundation Trust and UCL Great Ormond Street Institute of Child Health is made possible by the NIHR Great Ormond Street Hospital Biomedical Research Centre. The views expressed are those of the author(s) and not necessarily those of the NHS, the NIHR or the Department of Health.

## Author Contributions

**Conceptualization:** Adriana Vázquez-Vázquez, Emeline Rougeaux, Sarah Dib, Mary Fewtrell, Jonathan C. Wells.

**Formal analysis:** Adriana Vázquez-Vázquez.

**Investigation:** Adriana Vázquez-Vázquez, Emeline Rougeaux, Sarah Dib, Mary Fewtrell, Jonathan C. Wells.

**Methodology:** Adriana Vázquez-Vázquez, Emeline Rougeaux, Sarah Dib, Mary Fewtrell, Jonathan C. Wells.

**Supervision:** Mary Fewtrell, Jonathan C. Wells.

**Visualization:** Adriana Vázquez-Vázquez.

**Writing – original draft:** Adriana Vázquez-Vázquez.

**Writing – review & editing:** Adriana Vázquez-Vázquez, Emeline Rougeaux, Sarah Dib, Mary Fewtrell, Jonathan C. Wells.

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
