## [Decision Letter · Decision Letter 0]

3 Aug 2023

PGPH-D-23-01057

The impact of maternal mood and economic stress during Covid-19 lockdown on infant behaviour: findings from the UK Covid-19 New Mum Study

Dear Dr. Vazquez Vazquez,

Thank you for submitting your manuscript to PLOS Global Public Health. After careful consideration, we feel that it has merit but does not fully meet PLOS Global Public Health’s publication criteria as it currently stands. Therefore, we invite you to submit a revised version of the manuscript that addresses the points raised during the review process.

Please see the comments from two reviewers below. Both reviewers have raised some queries about the study design and motivations, which we now invite you to address. Please note that reviewer 2's final comment about figure quality does not need addressing - this is an artefact of the PDF generation, and will be fine for eventual publication. Please also note that there is no requirement to cite any specific references suggested by the reviewers.

We look forward to receiving your revised manuscript.

Kind regards,

Hanna Landenmark

Staff Editor

Journal Requirements:

1. Please provide additional details regarding participant consent. In the ethics statement in the Methods and online submission information, please ensure that you have specified (1) whether consent was informed and (2) what type you obtained (for instance, written or verbal, and if verbal, how it was documented and witnessed). If your study included minors, state whether you obtained consent from parents or guardians. If the need for consent was waived by the ethics committee, please include this information.

2. Please provide separate figure files in .tif or .eps format.

Additional Editor Comments (if provided):

Reviewers' comments:

Reviewer's Responses to Questions

**Comments to the Author**

1. Does this manuscript meet PLOS Global Public Health’s publication criteria? Is the manuscript technically sound, and do the data support the conclusions? The manuscript must describe methodologically and ethically rigorous research with conclusions that are appropriately drawn based on the data presented.

Reviewer #1: No

Reviewer #2: Partly

2. Has the statistical analysis been performed appropriately and rigorously?

Reviewer #1: Yes

Reviewer #2: No

3. Have the authors made all data underlying the findings in their manuscript fully available (please refer to the Data Availability Statement at the start of the manuscript PDF file)?

Reviewer #1: Yes

Reviewer #2: No

4. Is the manuscript presented in an intelligible fashion and written in standard English?

Reviewer #1: Yes

Reviewer #2: Yes

5. Review Comments to the Author

Reviewer #1: This study investigates the impact of maternal mood and economic stress during the early stages of the COVID19 pandemic on infant crying and fussiness in a sample of 2031 mother-infant dyads. Main results show that maternal wellbeing plays a significant role in children’s behavioural changes during the pandemic. Although the topic of the paper is interesting and the sample size is big, there are significant methodological issues that severely affect the quality of this study.

Main concerns

- Data collection covers a wide time frame, from May 2020 to June (February?) 2021. Consequently, it is difficult to distinguish the effects of lockdown restrictions from those related to less stressful pandemic period on both maternal mental health and infant’s behaviour.

- Mothers were asked to evaluate changes in behaviour of their infants as a result of staying home or the COVID-19 pandemic. Retrospective bias need to be considered, especially when participants were recruited one year after the outbreak of the COVID-19 pandemic.

- No validated instruments were used to assess maternal psychosocial wellbeing and infant crying and fussing.

Reviewer #2: Thanks for giving me the opportunity to read this paper, which investigates an important topic concerning the link between maternal mental health during the lockdown and infant’s behavior in a large sample of 2031 families from the UK Covid-19 New Mum Study. Results revealed an association between maternal mental health, coping and financial strain and changes in infant’s behavior during the lockdown (i.e. crying more and being fussier). The manuscript is clear and nicely written. The topic is interesting and important. The large sample collected during a challenging time is a strength of the current paper. However, I have some concerns that temper enthusiasm for publication of the manuscript in its current form.

Please find my major concerns better outlined below

1. Introduction: on page 3 it is stated that “the Covid-38 19 New Mum Study aimed to

capture information on maternal experiences and mood and infant feeding practices and

behaviour during the initial and subsequent stages of the lockdown”. Please specify whether prenatal or postnatal maternal experiences were the focus of the study and provide rationale for this. It might be important to refer also to available meta-analytical evidence showing that maternal mental health was significantly worsened from pre-pandemic period to during the pandemic (e.g. Tomfohr-Madsen et al., 2021)

2. On page 3 the authors reported “The impacts on maternal mental health may be of particular importance for early child development. Studies have suggested that maternal prenatal and postnatal stress could influence cognitive development and temperament in children (4–8) Since the Covid-19 outbreak, women have been exposed to a stressful situation of unknown duration that can be influencing their children’s behaviours (9–15).”. While it is important to mention this kind of broader literature, I think it is also important to acknowledge that a number of studied has already showed a link between maternal distress related to the pandemic and infant’s temperament and development, for example Provenzi et al., 2021; Nazzari et al., 2023, Giesbrecht et al., 2022).

3. I think it would be important to include in the introduction or method section a paragraph providing more context about the lockdown period in the UK in order to be able to better understand how it might have impact maternal and infant bhevaior as well as draw comparisons with data from other countries.

4. Methods: the questionnaires employed in the online survey are not clearly mentioned, so I am not sure whether any kind of standardized instrument has been employed. As several widely employed and well-validated questionnaires exists to assess maternal mood, coping and infant behavior, I would like to know whether the authors employed any of this or reasons for not doing so.

5. Results: the authors computed two principal component respectively for maternal mental health and coping, however in the analyses reported on page 10 and 11 they run univariable logistic regression including all items (I guess) of the survey as possible predictors of infant behavior. Thus the analytical strategy is not really entirely clear to me.

6. Please follow APA style to report statistical results both in text and tables (e.g., β, SE, p-values are missing)

7. Please provide model significance of multivariable logistic regression analysis.

8. Considering the large sample why did the authors did not choose to test any moderation/mediation model? For example, it might be interesting to test whether financial strain might further exacerbate the impact of maternal mental health on infant behavior as well as whether maternal coping might buffer the impact of adverse conditions (e.g. high financial stress or low maternal mood).

9. The authors stated that “Moreover, we did not formally assess maternal mental health and infant behaviour due to the anonymous nature of the survey which made it not possible to identify participants for follow up” – this sentence is not clear and need to be rephrased

10. Figures are of low quality and need to be graphically improved

6. PLOS authors have the option to publish the peer review history of their article (what does this mean?). If published, this will include your full peer review and any attached files.

**Do you want your identity to be public for this peer review?** For information about this choice, including consent withdrawal, please see our Privacy Policy.

Reviewer #1: No

Reviewer #2: No

---

## [Decision Letter · Decision Letter 1]

19 Mar 2024

The impact of maternal mood and economic stress during Covid-19 pandemic on infant behaviour: findings from the cross-sectional UK Covid-19 New Mum Study

PGPH-D-23-01057R1

Dear Miss Vazquez Vazquez,

We are pleased to inform you that your manuscript 'The impact of maternal mood and economic stress during Covid-19 pandemic on infant behaviour: findings from the cross-sectional UK Covid-19 New Mum Study' has been provisionally accepted for publication in PLOS Global Public Health.

Best regards,

Julio Croda, Ph.D, M.D.

Academic Editor

Reviewer Comments (if any, and for reference):

Reviewer's Responses to Questions

**Comments to the Author**

1. If the authors have adequately addressed your comments raised in a previous round of review and you feel that this manuscript is now acceptable for publication, you may indicate that here to bypass the “Comments to the Author” section, enter your conflict of interest statement in the “Confidential to Editor” section, and submit your "Accept" recommendation.

Reviewer #2: All comments have been addressed

Reviewer #3: All comments have been addressed

2. Does this manuscript meet PLOS Global Public Health’s publication criteria? Is the manuscript technically sound, and do the data support the conclusions? The manuscript must describe methodologically and ethically rigorous research with conclusions that are appropriately drawn based on the data presented.

Reviewer #2: Yes

Reviewer #3: Yes

3. Has the statistical analysis been performed appropriately and rigorously?

Reviewer #2: Yes

Reviewer #3: Yes

4. Have the authors made all data underlying the findings in their manuscript fully available (please refer to the Data Availability Statement at the start of the manuscript PDF file)?

Reviewer #2: Yes

Reviewer #3: Yes

5. Is the manuscript presented in an intelligible fashion and written in standard English?

Reviewer #2: Yes

Reviewer #3: Yes

6. Review Comments to the Author

Reviewer #2: (No Response)

Reviewer #3: I would like to express my gratitude for the opportunity to review your article. I am pleased to note that all suggestions and queries raised by previous reviewers have been addressed and resolved.

Your work opens up avenues for new studies and answers some questions about the impacts of the COVID-19 pandemic on modern life. The depth and breadth of your research are commendable, and the insights you provide are both timely and significant.

Furthermore, I would like to add that I do not have any new questions or suggestions at this time. The revisions made and the depth of the research presented have addressed all the points of consideration.

7. PLOS authors have the option to publish the peer review history of their article (what does this mean?). If published, this will include your full peer review and any attached files.

**Do you want your identity to be public for this peer review?** For information about this choice, including consent withdrawal, please see our Privacy Policy.

Reviewer #2: No

Reviewer #3: **Yes: **Roberto D. Oliveira
